# Hepatocellular Carcinoma Presenting Simultaneously with Echinococcal Cyst Mimicking a Single Liver Lesion in a Non-Cirrhotic Patient: A Case Report and Review of the Literature

**DOI:** 10.3390/diagnostics12071583

**Published:** 2022-06-29

**Authors:** Jelena Djokić Kovač, Milica Mitrović, Aleksandra Janković, Marko Andrejević, Aleksandar Bogdanović, Predrag Zdujić, Uroš Đinđić, Vladimir Dugalić

**Affiliations:** 1Center for Radiology and Magnetic Resonance Imaging, University Clinical Centre of Serbia, Pasterova No.2, 11000 Belgrade, Serbia; dr_milica@yahoo.com (M.M.); jankovicm.alex@gmail.com (A.J.); 2Faculty of Medicine, University of Belgrade, Dr Subotica No. 8, 11000 Belgrade, Serbia; aleksandarbogdanovic81@yahoo.com (A.B.); vanjadug@gmail.com (V.D.); 3Departament for Pathology, University Clinical Centre of Serbia, Pasterova No.2, 11000 Belgrade, Serbia; drmarkoandrejevic@gmail.com; 4Clinic for Digestive Surgery, University Clinical Centre of Serbia, Koste Todorovica Street, No.6, 11000 Belgrade, Serbia; pedjazdujic@yahoo.com (P.Z.); djindjicuros93@gmail.com (U.Đ.)

**Keywords:** cystic echinococcosis, hepatocellular carcinoma, coexistence, liver

## Abstract

A coexistance of liver cystic echinococcosis (CE) and hepatocellular carcinoma (HCC) is very rare. HCC is the leading cause of cancer-related mortality worldwide, while CE is a globally endemic zoonosis caused by the cestode tapeworm *Echinococcus granulosus*. The association between these two diseases is still not well-defined. A preoperative diagnosis may be challenging, especially if HCC and CE present as a single lesion and if atypical imaging features are present. Herein, we present a case of the patient that was initially diagnosed as an extensive necrotic tumor in the left liver lobe and highly suspicious of being HCC associated with peritumoral hematoma. Left hemihepatectomy was performed, and the histopathological findings showed the collision of two lesions: a hydatid cyst and HCC.

## 1. Introduction

Hepatocellular carcinoma (HCC) is the most common primary liver cancer and the fourth-most frequent malignancy in the general population [1]. Although the diagnosis of HCC is straightforward when a typical vascular pattern is observed in patients with cirrhotic liver, the diagnosis might be challenging in cases of atypical HCC [2]. In this regard, in addition to atypical vascular behaviour, cystic degeneration, intratumoral haemorrhage with tumor rupture and peritumoral hematoma, the presence of calcifications might complicate a preoperative diagnosis [3]. Human cystic echinococcocis (CE) or hydatid disease is an infectious parasitic disease caused by *Echinococcus granulosus* [4]. The liver is the most commonly affected organ, followed by the lungs and brain [5]. With the popularization of tourism and migration of the population, CE is no longer just an endemic disease, but it has become a global health problem worldwide [4]. Even though both HCC and liver echinococcosis are common diseases, their coexistence, especially in the form of a coalescent lesion in a non-cirrhotic liver, is very rare in clinical practice [6,7,8,9,10,11]. The association between these two diseases is still not well-defined, with many contradictory reports in the previous literature [12,13,14]. Herein, we present a case of the synchronous occurrence of HCC and a hydatid cyst mimicking a single lesion on preoperative imaging.

## 2. Case Report

A 58 year-old female patient was admitted to our hospital (University Clinical Center of Serbia, Belgrade) because of persistent epigastric pain, which lasted one month. The accompanying symptoms included weight loss and nausea. Her previous medical history did not indicate chronic liver disease, nor were other comorbidities present. On admission, an abdominal ultrasound examination revealed a large, solid, cystic lesion occupying the left liver lobe. Routine laboratory tests, including liver function analyses and tumor markers, were within normal limits. Serological tests for the viral hepatitis were negative. The patient underwent computed tomography (CT) examination of the abdomen, which showed a large, predominantly cystic lesion measuring up to 14 cm in diameter occupying lateral part of the left liver lobe with extrahepatic growth in the gastrohepatic ligament, compressing the stomach. The walls of the cyst were thick, irregular with coarse calcifications (Figure 1A). Adjacent to the cystic lesion in liver segment IV, an ill-defined solid component in the form of a multinodular tumor was detected, which showed arterial-phase enhancement and slight wash-out in the portal-venous phase (Figure 1B,C).

In order to further characterize the lesion, the patient was refered to a magnetic resonance imaging (MRI) examination. During the MRI, a clearly demarcated heterogeneous cystic lesion located in liver segments II and III and in the gastrohepatic ligament was seen. The internal content of the cyst displayed a heterogeneous signal intensity on the T2-weighted image with a central hyperintensity. Moreover, irregular, thick areas of hypointensity in the periphery were noted, which showed a high signal intensity in the T1-weighted image, indicating necrotic debris, and suggested the presence of internal hemorrhaging (Figure 2A,B). Next to the lateral contour of the cystic component, the solid part of the lesion was detected in liver segment IV. This part of the lesion displayed a slight hyperintensity in the T2-weighted image, restricted the diffusion in the diffusion-weighted images (DWI) and showed an intense arterial enhancement with subsequent washout in the portal-venous phase (Figure 2C,D and Figure 3). On the basis of the above-mentioned imaging findings, the diagnosis of multinodular HCC complicated with hemorrhagic transformation and peritumoral hematoma development was suspected. The background liver parenchyma appeared normal without CT and MRI signs of a cirrhotic configuration. Moreover, there was no dissemination in the other organs. 

The patient underwent left hemihepatectomy, and the tumor was completely resected, with negative margins. The macroscopic findings clearly showed that the cystic portion of the tumor corresponded to a parasitic cyst adjacent to the malignant alteration of the liver parenchyma (Figure 4). A further histological examination revealed the collision of two lesions: an echinococcal cyst and well-differentiated HCC (Figure 5). The postoperative course was uneventful, and the patient was dismissed from the hospital ten days after surgery. Following the WHO-Informal Working Group on Echinococcosis (WHO-IWGE) recommendations, antiparasitic treatment after surgery was not indicated. The patient underwent follow-up MRI examinations every three months after surgery for one year and up to now, has shown no recurrence of the disease.

## 3. Discussion

Although both HCC and CE are common liver lesions with high incidence rates in the general population, their synchronous occurrence is very rare in clinical practice, especially in a non-cirrhotic liver [6,7,8,9]. Thus, in a recent study among 3300 patients with CE, the concomitant presence of HCC was found in only 13 cases (0.39%) [8]. To our knowledge, there are 26 cases reported in the literature, with only four cases where HCC and CE presented as a single lesion [6,10,11]. Furthermore, in all previously reported cases where HCC and CE presented as a single lesion, CE had typical preoperative imaging findings and mostly presented as CE1 or CE2 lesions.

When HCC and CE mimick a single liver lesion, a preoperative diagnosis might be very challenging. If typical imaging features of CE are present in the form of cystic lesions with multiple internal daughter cysts and the absence of postcontrast enhancement, the diagnosis of cystic echinococcosis is straightforward [15]. However, in coexisting cases, inactive hydatid lesions that can imitate liver tumors might be seen [8]. Similar to previous reports, our patient had a CE4 hydatid cyst, which closely resembled a necrotic liver tumor. Moreover, since the internal content of CE may have a high signal intensity in T1-weighted images, the diagnosis of hemorrhagic degeneration in solid tumors might be suggested. In such cases, the absence of postcontrast enhancement favors the diagnosis of liver echinococcosis [15]. Nevertheless, the presence of a solid lesion demonstrating a typical vascular pattern for HCC adjacent to the cystic lesion with internal T1-weighted hyperintensity in the present case was highy suspicious of HCC with hemorrhagic transformation and the development of peritumoral hematoma. Ruptured HCC is a rare complication of HCC and is seen in up to 15% of HCC patients [16]. Typical imaging features of ruptured HCC include hemoperitoneum, perihepatic hematoma, liver tumor associated with discontinuity of the liver surface and enucleation sign [17]. A useful tip for the differentiation of hydatid disease from hemorrhagic liver tumors is the visualization of dystrophic calcifications [18]. As the detection of calcifications is difficult in a MRI examination, an additional CT examination is recommended in all problematic cases. Although HCC may rarely be partly calcified, the distribution and morphology of the calcification, especially if they are seen in thick necrotic debris, should indicate a degenerated, inactive form of CE [19]. The difficulties in a preoperative diagnosis in patients with concomitant HCC and CE were also pointed out in previous studies. Thus, in the study by Bo et al., among 13 cases, only one patient was correctly diagnosed preoperatively [8]. 

The treatment for HCC depends on the stage of the disease. The BCLC staging system is a currently widely used method to stage HCC and provides guidelines for treatment [20]. According to BCLC, our patient was considered as early-stage HCC (Stage A) and was treated successfully with a left hemihepatectomy, since there were no clinical comorbidities. The suggested stage-specific approach for uncomplicated cystic echinococcosis of the liver, stage CE4 and CE5, is “watch and wait” [21]. However, if a cyst exerts pressure on adjacent organs, surgery is recommended. Even though the cyst in our patient was not initially recognized as CE, since it was very large with extrahepatic growth, compressing the stomach, the left hemihepatectomy was the appropriate treatment. In the present case, in accordance with the WHO-IWGE recommendations, antiparasitic treatment following surgery was not indicated [21]. In patients with a preoperatively recognized synchronous occurrence of HCC and CE the treatment principles follow the Romic classification with hemihepatectomy recommended for the type 3a category, which included the patient from the present case [7]. 

There is an increasing number of studies indicating that there might be a possible connection between *Echinoccocus granulosus* infection and malignant tumor development [6,12,13]. The procancerogenic effect of an echinococcus infestation related to the modulation of the host immune response has been pointed out in a few stuides [6,12,13]. In recent research, in vitro and ex vitro studies have shown that *Echinoccocus granulosus* protoscoleces have a promoting effect on the growth, migration and invasion capacities of HCC cells [22]. If HCC develops adjacent to CE, it might be speculated that a chronic inflammatory reaction in the surrounding parenchyma induces HCC development [11,12]. However, in almost all reported cases of HCC and CE presenting as a single lesion, the background parenchyma was cirrhotic, indicating that the concomitant presence of HCC and CE is a coincidence [8]. Nevertheless, in cases where both lesions develop without underlying cirrhosis, as was in the present case, the procarcinogenic effect of CE might be possible. Thus, it was shown that E. multilocularis phosphoglucose isomerase displays a sequence similarity of 86% with human phosphoglucose isomerase, which may act as a growth factor and promote carcinogenesis, as was recently reported by Stadelmann et al. [23]. Nonetheless, the strong evidence supporting the fact that CE might induce mutations within hepatocyte DNA, and the subsequent carcinogenesis is still lacking. In contrast, there are a growing number of studies indicating the protective effect of CE on HCC progression with a prolonged overall survival time of HCC patients that had a synchronous CE infection [8,14]. Antigenic similarities between *Echinococcus granulosus* and some malignant tumors might be the possible mechanisms of echinococcal antitumor activity [24]. In this regard, few animal studies have shown that the serum of patients with CE had an antitumor activity on the growth of non-small-cell lung cancer, and also could suppress colon cancer progression [25,26]. Furthermore, there are also data showing that echinococcus could suppress tumor growth, inducing the proliferation of natural killer cells [27]. However, more studies are needed to clarify the possible connection between CE and HCC and to elucidate the underlying molecular mechanisms in patients without liver cirrhosis. 

## 4. Conclusions

The coexistence of HCC and CE as a solitary lesion on the grounds of a non-cirrhotic liver is extremely rare in clinical practice. If typical imaging features are absent, a preoperative diagnosis is very challenging. In such cases, the presence of calcifications inside the cystic lesion adjacent to the solid tumor should indicate the possibility of concomitant CE and HCC. Therefore, if there is any diagnostic dilemma, a serological test for hydatid disease should be performed. Based on the present case, it might be hypothesized that CE may promote the development of HCC in patients without underlying cirrhosis. Nevertheless, further studies are needed to clarify the possible relation among these diseases in patients without chronic liver disease.

## Figures and Tables

**Figure 1 diagnostics-12-01583-f001:**
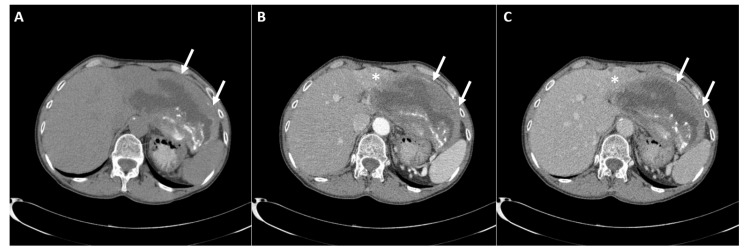
Computed tomography examination native scan (**A**) shows a large, heterogeneous, predominantly cystic lesion with irregular, eccentric, coarse calcifications inside necrotic debris (arrows). After the intravenous contrast administration arterial phase (**B**) shows an ill-defined hyper-vascularized solid tumor (asterisk) in liver segment IV adjacent to the cystic lesion (arrows). In the portal-venous phase (**C**) a discrete washout of the solid tumor is seen. Note the absent opacification of the internal cystic component, with only a slight enhancement of the cystic wall.

**Figure 2 diagnostics-12-01583-f002:**
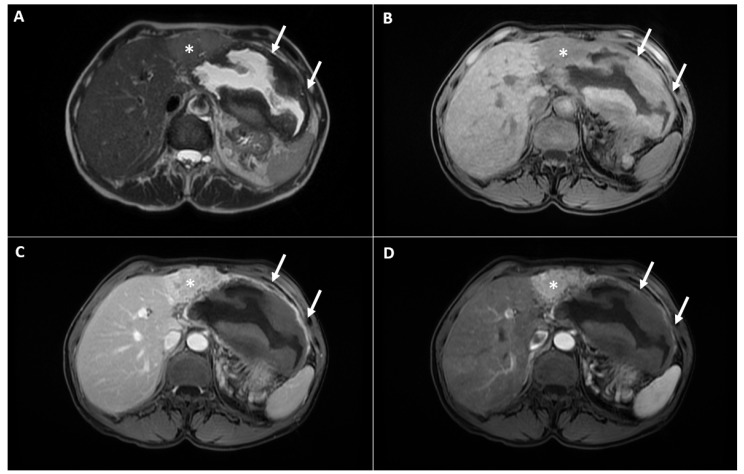
The MRI examination shows a cystic lesion with thickened, irregular walls (arrows) and a solid tumor in liver segment IV adjacent to the cystic component (asterisk) displaying an intermediate signal intensity in the T2-weighted image (**A**), a low signal intensity in the T1-weighted image (**B**) and an intense enhancement in the arterial phase (**C**), with washout in the portal-venous phase (**D**). Note only a slight enhancement of the cystic wall, while there is no enhancement in the necrotic masses along the walls of the cyst.

**Figure 3 diagnostics-12-01583-f003:**
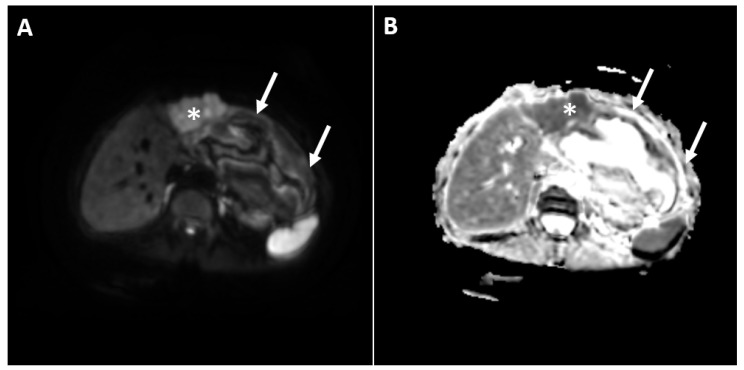
The diffusion weighted image (**A**) shows a solid tumor (asterisk) with a high signal intensity, indicating hypercellularity, and a cystic lesion next to the solid tumor (arrows) with a low signal intensity, suggesting acellularity. The corresponding ADC map is shown (**B**), with the ADC value of the solid tumor being 0.891 × 10^−3 ^mm^2^/s.

**Figure 4 diagnostics-12-01583-f004:**
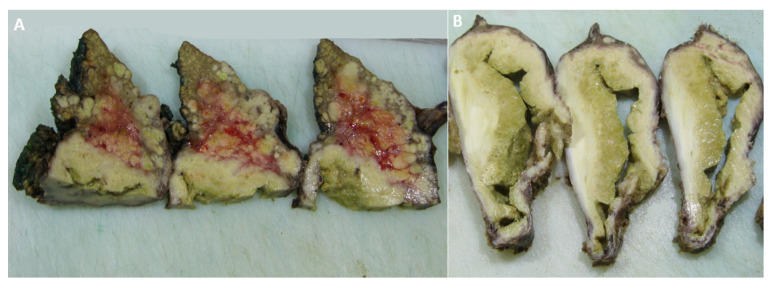
The macroscopic specimen shows a cross-section of the tumor tissue corresponding to the malignant alteration of the liver parenchyma (**A**) and the parasitic cyst next to the tumor (**B**).

**Figure 5 diagnostics-12-01583-f005:**
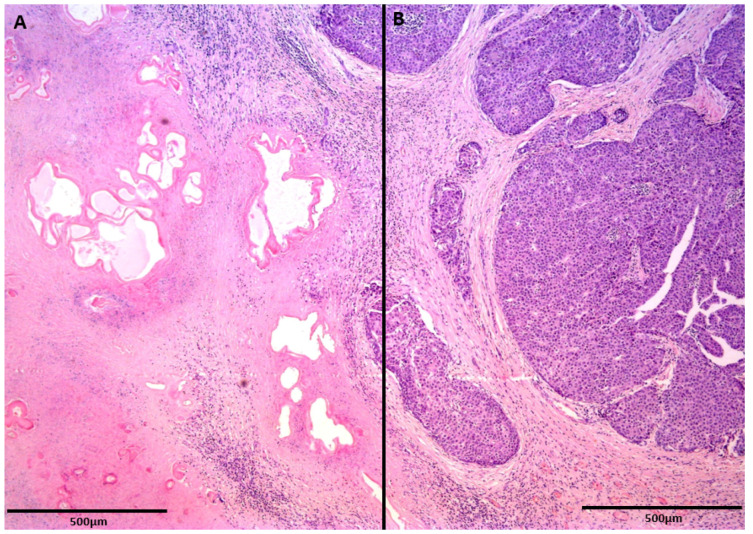
A histological examination clearly depicts the collision of two lesions: parasitic cysts surrounded by a fibrohistiocytic rim (**A**) and peripheral nodular and trabecular proliferation of hepatocellular carcinoma (**B**). Scale bar = 500 µm.

## Data Availability

Not applicable.

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
