# Peer review of "Hepatocellular Carcinoma Presenting Simultaneously with Echinococcal Cyst Mimicking a Single Liver Lesion in a Non-Cirrhotic Patient: A Case Report and Review of the Literature"

_diagnostics, 2022, doi:10.3390/diagnostics12071583_

Round 1

Reviewer 1 Report

To editors and authors

Hepatocellular carcinoma presenting simultaneously with echinococcal cyst mimicking a single liver lesion in a non-cirrhotic patient: a case report and review of the literature

This is a very interesting manuscript that should be considered for publication in DIAGNOSTICS after some revisions below.

1) Please recheck and revise cautiously citation and references as MDPI format.

2) Arrows in figures are very small.

3) Add scale bars in histopathological images.

4) Please show ADC value in the figure legends.

5) Figure 5 needs to have icons to indicate clearly two lesions along with appropriate legend.

Sincerely

Author Response

Response to Reviewer 1 Comments

Point 1: Please recheck and revise cautiously citation and references as MDPI format.

Response 1: All references were cautiously checked, and a few errors were found which were corrected according to MDPI guidelines.

Point 2: Arrows in figures are very small.

Response 2: Corrected in revised manuscript.

Point 3: Add scale bars in histopathological images.

Response 3: Corrected in revised manuscript (Pg:5, Ln:3).

Point 4: Please show ADC value in the figure legends.

Response 4: Corrected in revised manuscript (Pg:4, Ln:4).

Point 5: Figure 5 needs to have icons to indicate clearly two lesions along with appropriate legend.

Response 5: Corrected in revised manuscript.

Reviewer 2 Report

This study is well designed in terms of content and structure. The authors presented a case of synchronous occurrence of HCC and hydatid cyst mimicking a single lesion on preoperative imaging.

After reviewing the manuscript, I have minor comments.

1.   In line 50, mention the hospital and the city where the patient was admitted.

2. In the case report section, describe the patient's treatment process after surgery.

3. In the discussion section, it is better to describe whether cystic echinococcosis promote the development of hepatocellular carcinoma or have an antitumor role. So far, several studies have been conducted on the relationship between parasites (especially Echinococcus granulosus) and cancer. Therefore, it is necessary to discuss the mechanisms in more detail.

Author Response

Response to Reviewer 2 Comments

Point 1: In line 50, mention the hospital and the city where the patient was admitted.

Response 1: Added in revised manuscript (Pg:2, ln:9-10).

Point 2: In the case report section, describe the patient's treatment process after surgery.

Response 2: Added in revised manuscript (Pg:4, ln:12-14).

Point 3: In the discussion section, it is better to describe whether cystic echinococcosis promote the development of hepatocellular carcinoma or have an antitumor role. So far, several studies have been conducted on the relationship between parasites (especially Echinococcus granulosus) and cancer. Therefore, it is necessary to discuss the mechanisms in more detail.

Response 3: The relationship between Echinococcus granulosis infection and HCC development has been thoroughly discussed in the discussion section. The possible pro-cancerogenic effect of echinococcus infestation was explained according to the results of the recent studies. Moreover, the studies indicating protective effect of CE on HCC progression and possible mechanism of echinococcal anti-tumor activity were also discussed. Please find details in revised manuscript Pg:6, ln:29-43, and Pg:7, ln:1-12.

Reviewer 3 Report

In the manuscript titled “Hepatocellular carcinoma presenting simultaneously with echinococcal cyst mimicking a single liver lesion in a noncirrhotic patient: a case report and review of the literature” Jelena Djokic Kovač et al. present a case report on patient diagnosed with hepatocellular carcinoma. The patient underwent a liver resection and final histopathological findings showed hepatocellular carcinoma and echinococcal cyst. The case is interesting, and the manuscript is well written. However, I still have some comments:

1.     It is not clear if serological test for hydatid disease were performed.

2.     Was cirrhosis and liver function before surgery evaluated? How long was the follow-up period?

3.     After reading the manuscript, I’m not sure what is the main message of it. Some recommendations might be included in the conclusion. 

Author Response

Response to Reviewer 3 Comments

Point 1: is not clear if serological tests for hydatid disease were performed.

Response 1: Serological tests for hydatid disease were not performed since we did not suspect cystic echinococcosis liver infection preoperatively. Rather we suspected haemorrhagic transformation of hepatocelullar carcinoma associated with peritumoral hematoma.

 Point 2: Was cirrhosis and liver function before surgery evaluated? How long was the follow-up period?

Response 2: The liver function analyses were within normal limits. Serological tests for the viral hepatitis were negative. The patient had no history of previous chronic liver disease, nor alcohol intake. The follow-up period is one year since the patient was operated last year, and is still without signs of disease recurrence.

Point 3: After reading the manuscript, I’m not sure what is the main message of it. Some recommendations might be included in the conclusion. 

Response 3: Corrected in revised manuscript (Pg:7, ln:16-19).

Round 2

Reviewer 3 Report

-